# Sensor Based on PZT Ceramic Resonator with Lateral Electric Field for Immunodetectionof Bacteria in the Conducting Aquatic Environment [note 1]

**DOI:** 10.3390/s20103003

**Published:** 2020-05-25

**Authors:** Irina Borodina, Boris Zaitsev, Andrey Teplykh, Gennady Burygin, Olga Guliy

**Affiliations:** 1Kotelnikov Institute of Radio Engineering and Electronics of RAS, Saratov Branch, 410019 Saratov, Russia; borodinaia@yandex.ru (I.B.); teplykhaa@mail.ru (A.T.); 2Institute of Biochemistry and Physiology of Plants and Microorganisms, Russian Academy of Sciences, 410049 Saratov, Russia; burygingl@gmail.com (G.B.); guliy_olga@mail.ru (O.G.)

**Keywords:** immunodetection of microbial cells in aqueous medium, PZT resonator with lateral electric field, electrical impedance, antibodies (Abs)

## Abstract

A biological sensor for detection and identification of bacterial cells, including a resonator with a lateral electric field based on PZT ceramics was experimentally investigated. For bacterial immunodetection the frequency dependencies of the electric impedance of the sensor with a suspension of microbial cells were measured before and after adding the specific antibodies. It was found that the addition of specific antibodies to a suspension of microbial cells led to a significant change in these frequency dependencies due to the increase in the conductivity of suspension. The analysis of microbial cells was carried out in aqueous solutions with a conductivity of 4.5–1000 μS/cm, as well as in the tap and drinking water. The detection limit of microbial cells was found to be 10^3^ cells/mL and the analysis time did not exceed 4 min. Experiments with non-specific antibodies were also carried out and it was shown that their addition to the cell suspension did not lead to a change in the analytical signal of the sensor. This confirms the ability to not only detect, but also identify bacterial cells in suspensions.

## 1. Introduction

The development of new biosensor systems for the detection and identification of microbial cells in aquatic environments is an urgent problem in modern microbiology. It is important to consider the possibility of analysis in conditions of increased conductivity of the measurement medium, since the high conductivity of the medium significantly complicates the process of detection and identification of bacteria. Most existing biosensors for detecting bacteria are poorly adapted for measurements in water used in industry, municipal, and commercial institutions, the electrical conductivity of which is high due to the presence of various ionic compounds [1,2,3,4,5,6].

Therefore, new methods are being actively developed for the analysis of microorganisms in conductive liquids, which are distinguished by the high sensitivity, accuracy, and speed. In this regard, the electro-acoustic analysis methods are promising [7,8,9]. For the detection and identification of microbial cells, biosensors based on a resonator with a lateral electric field are widely used [10,11,12,13,14,15,16,17,18]. Nevertheless, the problem of increasing the sensitivity of such sensors remains relevant.

Researchers have also paid attention to finding the promising new materials for acoustic biosensors. Traditionally, such materials as lithium tantalate, lithium niobate, and quartz have been widely used to develop sensors based on resonators with a lateral electric field [16,17,18]. However, the use of PZT piezoceramics for the development of acoustic sensor technology is also promising [19]. This material is characterized by a high coefficient of electromechanical coupling [20], i.e., the corresponding sensors will be more sensitive to changes in the conductivity of the contacting liquid. An additional advantage of piezoceramics is low cost, which opens up great opportunities for the mass production of sensors.

The possibility of using a sensor based on a resonator with a lateral electric field made of PZT ceramics for the express analysis of microbial cells in aqueous solutions with a conductivity of 4.5–1000 μS/cm, as well as in the tap and drinking water, was investigated.

## 2. Materials and Methods

### 2.1. The Microbial Cells and Reagents

*Azospirillum brasilense* strain Sp7 (IBPPM 150) and *Escherichia coli* strain K-12 (IBPPM 204) were taken from the IBPPM RAS Collection of Rhizosphere Microorganisms (http://collection.ibppm.ru/). Microbial cells were grown in Luria–Bertani’s liquid medium and prepared for measurement as described in [14]. Before analysis, the bacterial cells were thoroughly washed in distilled water by the centrifugation (three times, at 2800× *g*, 5 min each) and were re-suspended in distilled water (conductivity, 4.5 μS/cm). The absorbance of the prepared suspension was brought to *D*_600_ = 0.4–0.42. Then 1/10, 1/100, 1/1000, etc., dilutions (1 part of cell solution, 9 parts of buffer solution), were used.

The antibodies (Abs) specific to the lipopolysaccharide of *A. brasilense* Sp7 were grown as described in [21]. The strain-specificity of the antibodies was studied by the double immunodiffusion in the 1% agarose gel by using the standard technique [22].

### 2.2. Enzyme-Linked Immunosorbent Assay (ELISA)

Detection of interactions of antibodies with bacterial cells was carried out by ELISA in 96-well polystyrene plates using the standard procedure, as previously described in [23]. Aliquots (50 μL) of each bacterial suspension in twofold dilutions (initial concentration, 10^8^ cells/mL) were immobilized in the wells through simple adsorption, and kept for 30 min on a shaker at room temperature. The samples were replaced with 100 μL of 0.05% polyethylene glycol 20,000 (PEG), added to each well to block the free binding sites on polystyrene. This solution was replaced by 50 μL of Abs specific to the lipopolysaccharide of *A. brasilense* Sp7 (final conc. 0.2 mg/mL) diluted in phosphate-buffered saline (PBS) with 0.02% Tween 20 and 0.005% PEG (for prevention of nonspecific Ab sorption). After incubation for 30 min, the wells were washed three times with 100 μL of PBS–0.02% Tween 20. Horseradish peroxidase conjugated with goat anti-rabbit antibodies (Jackson ImmunnoResearch Laboratories, West Grove, PA, USA; diluted 1:2000) was used as an enzyme label. Peroxidase activity was estimated by adding to each well 50 μL of a substrate mixture of 0.03% *o*-phenylenediamine and 0.02% hydrogen peroxide in 0.1 M sodium citrate buffer (pH 4.5). The enzyme reaction was stopped with 100 μL of 1 N H_2_SO_4_. The absorbance at 492 nm was read on a Multiskan Ascent analyzer (Thermo, Vantaa, Finland). Absorbance values at 492 nm for the wells without bacterial cells were used as the control. Data were processed with Microsoft Excel 2003 software (Microsoft Corp., Redmond, USA); 95% confidence intervals are given.

### 2.3. Electron Microscopy

The samples for electron microscopy were prepared as described in [24] and were viewed by a Libra 120 transmission electron microscope (Carl Zeiss, Oberkochen, Germany) at an accelerating voltage of 120 kV. The microscopy of the samples was done at the Simbioz Center of the IBPPM RAS.

### 2.4. Measuring Medium

The analysis of microbial cells was carried out directly in the liquid phase. We used distilled water (the conductivity = 4.5 μS/cm) and standard 0.12 M phosphate-buffered saline (PBS) (pH 7.2) of the following composition (g/L): KH_2_PO_4_, 0.43; Na_2_HPO_4_, 1.68; NaCl, 7.2 (the conductivity = 10, 20, 50, 100, 300, 500, and 1000 μS/cm). The conductivity of the buffer was changed by adding distilled water and was determined with a HANNA HI 8733 (Ecoinstruments, Moscow, Russia). The viscosity of the buffer solutions with microbial cells was measured using a viscometer SV-10 (A&D Company, Limited, Tokyo, Japan).

### 2.5. Description of Sensor and Method of Experiment

The general scheme of the experiment is presented in Figure 1. We used a sensor based on the resonator with a lateral electric field made of PZT ceramics. Two aluminum electrodes were deposited on one side of the resonator. The other side was the bottom of the liquid container. The characteristics of this sensor are described in more detail in [25].

For research, the resonator was connected to the impedance analyzer E4990A (Keysight Technologies, Santa Rosa, CA, USA), and the frequency dependences of the real and imaginary parts of the electrical impedance were measured in the frequency range 50–300 kHz. First, the characteristics of a sensor with a container filled with distilled water or a buffer solution with a given conductivity were studied. At the next stage, distilled water or a buffer solution with microbial cells was placed in the container and the sensor readings were again recorded. After this, the specific or nonspecific antibodies (Abs) were added to the container with a cell suspension, and the parameters of the sensor were measured. Measurements were carried out for different cell number (10^2^–10^8^ cells/mL), as well as for different antibodies concentration (1–6 μg/mL) added to the cell suspension. The conductivity of distilled water was equal to 4.5 μS/cm, the conductivity of buffer solutions varied from 10 to 1000 μS/cm.

As an analytical signal, we used the change in the maximum value of the real part of the electric impedance of the sensor after adding the specific antibodies to the suspension of bacterial cells.

## 3. Results and Discussion

### 3.1. Measurements in Distilled Water

Measurements of the parameters of the sensor with an empty container showed the presence of three resonances in the frequency dependences of the real (R) (Figure 2a) and imaginary (X) (Figure 2b) parts of the electrical impedance of the resonator. These resonant peaks were observed near the frequencies of 68.7, 97.8, and 264 kHz, and had the values of electromechanical coupling coefficient of 15%, 14%, and 5%, respectively [25].

In the next step, distilled water was added to the container. The addition of distilled water with a conductivity of 4.5 μS/cm led to the significant decrease in the maximum value of the real part of the electrical impedance (R_max_) for all resonant peaks in comparison with an empty container. This is apparently due to the fact that, firstly, the liquid layer exerted a massive load on the resonator. Secondly, the acoustic oscillations excited in this resonator were not purely shear, but had a small normal displacement component, which led to small radiation attenuation.

It was found that the quality factor of each resonance in contact with the liquid turned out to be significantly higher than one of a resonator with a longitudinal acoustic wave based on lithium niobate [13,14,15]. This is due to the fact that in a piezoceramic resonator the shear component of the mechanical displacement is prevalent, which significantly reduces the radiation loss of the acoustic wave when the surface of the resonator contacts the liquid [25].

Then, microbial cells, re-suspended in distilled water with a given concentration (10^2^–10^8^ cell/mL), were added to the container. It was shown that the addition of distilled water with microbial cells in container had practically no effect on the frequency dependences of the electrical impedance of resonator in comparison with adding the water without cells.

The surface of a microbe cell is a set of complicated antigen complexes. The distinctive feature of antibodies to form specific compounds with the appropriate antigens is used for determination of microorganisms. The same principle is employed to identify the microorganisms with the help of biosensor systems by using the Abs. Therefore, as a selective agent for determining bacteria, we used antibodies specific to them.

In the next measurement step the specific Abs were introduced into the cell suspension of *A. brasilense Sp7*. Different amounts of microbial cells (10^2^, 10^3^, 10^4^, 10^6^ and 10^8^, cells/mL) and specific Abs (1, 2, 4, and 6 μg/mL) were used in the experiments. The addition of specific Abs to *A. brasilense Sp7* led to a significant decrease in the real and imaginary parts of the electrical impedance of the resonator for almost all values of the studied cell concentrations excepting 10^2^ cells/mL.

As already noted, as an informative parameter, we used the maximum value of the real part of the electric impedance of the resonator because it was shown earlier that this parameter is preferable for measuring the conductivity and viscosity of a liquid [17].

Figure 3 presents, as an example, the frequency dependencies of the real part of the electric impedance of the sensor for cell suspension of *A. brasilense* Sp7 with concentration of 10^3^ cells/mL before (curve 1) and after (curve 2) adding the specific Abs with the amount of 4 μg/mL. The dependencies are presented for three resonance peaks near the frequencies: (a) 68.7 kHz, (b) 97.8 kHz, and (c) 264 kHz.

One can see that the addition of specific Abs to the suspension of microbial cells leads to a decrease in the maximum value of the real part of the electrical impedance (R_max_) by 63.5, 65.6, and 4.1 kOhm for resonance peaks near frequencies of 68.7, 97.8, and 264 kHz, respectively.

The changes in the parameters of the sensor can be explained as follows. When specific antibodies are added to the cell suspension, the active center of the antibody (Ab) joins with the antigenic (Ag) determinant. This highly specific process proceeds in aqueous solutions at a high speed. O-antigen of the Gram-negative bacteria is associated with lipopolysaccharide LPS of the cell wall. The determinant groups of this complex antigen are the terminal repeating units of polysaccharide chains attached to its main part. At the moment of binding the antibody to the determinant group of the antigen, a conformational rearrangement of the active center of the antibody occurs and the active antibody adapts to the antigen. The binding of antibodies to antigenic determinants on the cell surface is recorded by changing the polarization properties of the Ag-Ab complex. The interaction of an antigen molecule with an antibody or its active Fab fragment is accompanied, in turn, by changes in the spatial structure of the antigen molecule. This affects the change in the integrity of bacterial membranes, and leads to the release of cell contents into suspension and to the increase in its electrical conductivity. This increase in conductivity is recorded by the acoustic sensor as a decrease in the real and imaginary parts of the electrical impedance of the resonator near the resonant frequencies.

To exclude the nonspecific interactions, we used *E. coli* K-12 cells and Abs specific for *A. brasilense* Sp7 cells. Figure 4 shows, as an example, the frequency dependencies of the real part of the electric impedance of the sensor for a suspension of *E. coli* K-12 cells (10^3^ cells/mL) before (curve 1) and after (curve 2) the addition of Abs specific for *A. brasilense* Sp7 (4 μg/mL). These dependencies are presented for three resonance peaks near frequencies: (a) 68.7 kHz, (b) 97.8 kHz, and (c) 264 kHz. It can be seen that in this case no changes in the sensor parameters are observed.

To determine the specificity of Abs to the O-antigen of *A. brasilense* Sp7, an immunodiffusion analysis and enzyme-linked immunosorbent assay (ELISA) were performed. During double immunodiffusion, the specific antigen and antibody placed into wells of the agar gel diffuse towards each other and form precipitation in the form of a band at the zone of their meeting. According to the results of immunodiffusion, one can clearly see (Figure 5a) that Abs interacted only with the extract of *A. brasilense* Sp7 cells (formation of the antigen–antibody complex immunoprecipitation as the band in agarose gel) and did not interact with the extract of *E. coli* K-12 (no immunoprecipitation band). Thus, the Abs against O-antigen of *A. brasilense* Sp7 specifically formed the insoluble antigen–antibody complex only with the surface antigens of *A. brasilense* Sp7 [26] and did not form it with antigens of *E. coli* K-12. ELISA is characterized by the unique specificity of the immunochemical reaction, i.e., antibodies bind exclusively to certain antigens and are highly sensitive to antigen detection. The direct specificity of the Abs used in the work was also determined by ELISA.

The data presented in Figure 5b shows that Abs detected *A. brasilense* Sp7 cells, but did not detect *E. coli* K-12. These results indicate the absence of specific antigenic determinants on the cell surface of *E. coli* K-12.

Additionally, the electron microscopy of microbial cells was performed during their interaction with specific and nonspecific antibodies. Figure 6a presents the electron microscopic identification of *A. brasilense* Sp7 with Abs specific to *A. brasilense* Sp7 labeled with colloidal gold. Electron micrographs show that the markers of accumulation occur on the entire surface of the cells. Figure 6b shows a photograph of *E. coli* K-12 with Abs specific to *A. brasilense* Sp7. One can see that in this case no accumulation of antibodies around the cell is observed.

Figure 7 shows the time dependences of the maximum value of the real part of the resonator electrical impedance (R_max_) for resonant peaks near 68.7, 97.8, and 264 kHz for a specific interaction “*A. brasilense* Sp7—specific Abs”(a) and non-specific interaction “*E. coli* K-12—Abs specific for *A. brasilense* Sp7”(b).

At the initial time, a cell suspension was introduced into the container, and sensor readings were taken at intervals of 15 s. One could see no changes in the value of R_max_. Then (at 100th second), a specific/nonspecific reagent was introduced into the container with the cell suspension and the sensor readings were again recorded. It is seen that in the case of a specific interaction (Figure 7a), the value of R_max_ decreases sharply, and then the saturation process takes place. In the case of nonspecific interaction (Figure 7b), no changes in the sensor parameters occurred. The analysis time in all cases did not exceed 4 min.

We also performed the control experiments when Abs, specific to *A. brasilense* Sp7, were added to a container with water in the absence of microbial cells. Figure 8 shows the time dependencies of the real part of the electrical impedance of the resonator for the resonance peaks near frequencies 68.7, 97.8, and 264 kHz when Abs (4 μg/mL) specific for *A. brasilense* Sp7 cells were added to the container with water. It is seen that the addition of antibodies to water at the absence of cells did not lead to a change in the electrical impedance of the sensor.

Based on the measured frequency dependencies of the real part of the electrical impedance of the sensor, the dependencies of the change in the maximum value of the real part of the electrical impedance (ΔR_max_) on the cell concentration was constructed for a fixed value of the number of specific antibodies added to the cell suspension. Figure 9 shows, as an example, the dependencies of ΔR_max_ on the concentration of *A. brasilense* Sp7 cells when specific antibodies were added to the container with a cell suspension in an amount of 4 μg/mL. Dependences are presented for resonance peaks near frequencies 68.7, 97.8, and 264 kHz. It can be seen that the addition of specific Abs to a suspension of cells with a concentration of 10^2^ cells/mL practically does not change the R_max_ value (ΔR_max_ = 0). This can be explained by the fact that in this case the concentration of cells in the water is very low and the sensor cannot detect a change in the conductivity of the cell suspension due to the specific biological interaction of microbial cells with antibodies.

Significant changes in ΔR_max_ are observed when specific antibodies are added to a cell suspension with concentrations of 10^3^–10^8^ cells/mL. The variation range ΔR_max_ in these cases is 30–60 kOhm for resonance peaks near frequencies of 68.7 and 97.8 kHz, and 4–6 kOhm for a peak near the frequency of 264 kHz. Figure 9 shows that for cell suspensions with a concentration of 10^8^ cells/mL, a decrease in ΔR_max_ is observed compared with a concentration of 10^3^–10^6^ cells/mL for resonance peaks near the frequencies of 68.7 and 97.8 kHz. This is because the addition of a cell suspension with a concentration of 10^8^ cells/mL to a liquid container leads to some decrease in sensor signal even before the addition of specific Abs.

To explain this fact, we measured the viscosity and conductivity of cell suspensions with different concentrations and compared them with the viscosity and conductivity of distilled water before adding the cells. It was found that the viscosity of the cell suspension with any concentration of cells was equal to the viscosity of distilled water and amounted to 0.77–0.78 mPa s. With regard to the conductivity, the addition of cells with a concentration of 10^2^–10^6^ cells/mL to distilled water did not lead to a change in the conductivity of the cell suspension compared with the conductivity of distilled water. However, the addition of the cells with a concentration of 10^8^ cells/mL to distilled water led to an increase in conductivity from 4.5 μS/cm to 6.0 μS/cm. This can be explained by the fact that a high-frequency current (68–264 kHz) can pass not only through the space between the cells, but also through the cells.

Thus, we can say that the lower limit of determination of microbial cells using the studied sensor based on PZT ceramics is 10^3^ cells/mL. This value is an order of magnitude higher than the detection limit of the sensor based on lithium niobite [14,15].

The dependencies of a change in the maximum value of the real part of the electrical impedance (ΔR_max_) on the number of specific Abs (from 1 to 6 μg/mL) added to the cell suspension with a given value of their concentration were also constructed. Figure 10 shows, as an example, the dependencies of ΔR_max_ on the concentration of specific Abs added to a suspension of *A. brasilense* Sp7 cells with the concentrations 10^3^ cells/mL (Figure 10a) and 10^8^ cells/mL (Figure 10b) for three resonance peaks near frequencies 68.7, 97.8, and 264 kHz.

Figure 10a shows, that adding 1 μg/mL of antibodies to a cell suspension with a concentration of 10^3^ cells/mL leads to a change in R_max_ (ΔR_max_) of 42, 45, and 4.5 kOhm for resonances near the frequencies of 68.7, 97.8, and 264 kHz, respectively. The similar ΔR_max_ values are equal to 57, 65, and 6 kOhm for the quantity of antibodies of 5 μg/mL. Figure 10b presents the same data for a cell suspension with a concentration of 10^8^ cells/mL. In this case, the ΔR_max_ values were 32, 35, and 3 kOhm (the quantity of antibodies = 1 μg/mL) and 45, 50, and 5 kOhm (the quantity of antibodies = 5 μg/mL) for resonances near 68.7, 97.8, and 264 kHz, respectively. One can see a decrease in the sensor response for the concentration of 10^8^ cells/mL as compared to the concentration of 10^3^ cells/mL. An explanation of this fact is given after the commentary on Figure 9.

Thus, it was shown that the addition of specific Abs to the cell suspension leads to a significant change in R_max_ even with their minimal amount. Similar dependencies were constructed for other concentrations of microbial cells.

### 3.2. The Measurements in Buffers with Different Conductivity

As already noted, an important point is the ability to analyze cells in real liquids having high electrical conductivity (tap and industrial water, water in reservoirs, etc.). Therefore, in this section, we studied microbial cells in buffer solutions with different conductivities when interacting with antibodies using an acoustic sensor based on a resonator with a lateral electric field made of PZT ceramics. To carry out this measurement cycle, a measurement methodology was used similar to the procedure described in Section 2. At the first stage, buffer solution with a given conductivity was added to the liquid container and the sensor readings were measured. Then, the container was cleaned and dried, and a microbial buffer solution was added. After the stabilization of the sensor signal, we added Abs specific or nonspecific to the cells and again recorded the sensor readings. In this measurement cycle, *A. brasilense* Sp7 cells and *E. coli* cells with a concentration of 10^3^ cells/mL were used. The concentration of Abs was invariable and amounted to 6 μg/mL. The buffer solutions with a conductivity of 10, 20, 50, 100, 300, 500, and 1000 μS/cm were used.

As a result of the experiments, the frequency dependencies of the real part of the electrical impedance of the resonator were obtained for all values of the conductivity of the buffer solutions. It was shown that with an increase in the conductivity of the solution, the value of the real part of the electric impedance of the resonator decreased. These data are in good agreement with the data obtained earlier in [25], where the influence of liquid conductivity on the characteristics of a resonator based on PZT ceramics was studied. Adding microbial cells to the buffer solution, as in experiments with distilled water, did not affect the performance of the sensor. Subsequent addition of specific antibodies to the cell suspension led to a significant decrease in the real part of the electrical impedance of the sensor.

Figure 11 shows, as an example, the frequency dependencies of the real part of the electric impedance of the sensor filled by buffer solution with the conductivities of 10 μS/cm (Figure 11a,c,e) and 50 μS/cm (Figure 11b,d,f) with the cells *A. brasilense* Sp7 before (curve 1) and after (curve 2) adding the specific Abs with concentration of 6 μg/mL. The cell concentration is 10^3^ cells/mL.

One can see that the addition of specific Abs to a buffer solution with a conductivity of 10 μS/cm and microbial cells *A. brasilense* Sp7 (Figure 11a,c,e) leads to a decrease in the maximum value of the real part of the electrical impedance R_max_ by 32.1, 36.4, and 2.8 kOhm for resonant peaks near frequencies 67.8, 98.7, and 264 kHz, respectively. For a buffer solution with a conductivity of 50 μS/cm containing cells *A. brasilense* Sp7, the maximum value of R_max_ decreases by 11.3, 12.4, and 1.5 kOhm due to the addition of specific Abs for the above-mentioned resonance peaks (Figure 11b,d,f).

Based on the frequency dependencies of the real part of the electrical impedance of the sensor presented in Figure 11, the change in ΔR_max_ due to the addition of the specific Abs as function of the conductivity of the buffer solution with cells *A. brasilense* Sp7 were constructed. Figure 12 shows these dependencies for resonance peaks near frequencies 67.8, 98.7, and 264 kHz.

It can be seen that the value of ΔR_max_ decreases with increasing conductivity of the buffer solutions and reaches zero for the conductivity of 1000 μS/cm for all observed resonance peaks.

Table 1 shows the values of ΔR_max_ for suspensions of *A. brasilense* Sp7 cells with a concentration of 10^3^ cells/mL based on distilled water (4.5 μS/cm) and buffer solutions (10–1000 μS/cm) due to the addition of specific Abs in an amount of 6 μg/mL for resonant peaks near frequencies of 67.8, 98.7, and 264 kHz.

From the data given in the Table 1 it can be seen that the sensor can successfully detect specific reactions of microbial cells with Abs in solutions with conductivity up to 300 μS/cm. For the solutions with the higher conductivity, the measurement data are comparable with the measurement error. Additionally, the measurements were carried out with the samples of tap water (conductivity = 1000 μS/cm), as well as with the samples of purified drinking water (conductivity = 500 μS/cm). The data obtained are presented in the lower two rows of Table 1. They are well comparable with the data for buffer solutions.

As the control experiments, the interactions of a cell suspension with non-specific reagents were investigated. For this purpose, we used *E. coli* K-12 cells and Abs specific for *A. brasilense* Sp7 cells and non-specific for *E. coli* K-12 cells. Figure 13 shows the dependencies of ΔR_max_ on the conductivity of the buffer solution when Abs specific for *A. brasilense* Sp7 cells was added to the buffer with *E. coli* K-12 cells for resonance peaks near frequencies 67.8, 98.7, and 264 kHz.

One can see that the nonspecific interactions practically did not lead to a change in the parameters of the sensor for each value of the conductivity of buffer solution. Thus, a sensor based on a resonator with a lateral electric field made of PZT ceramics confidently distinguishes specific biological interactions from non-specific interactions. This means that the sensor can be used not only for the detection of microbial cells in the aquatic environment, but also for their identification.

It should be noted that in most works devoted to acoustic biological sensors [7,10,12], active layers with immobilized antibodies are used. Such an approach is distinguished, first, by the long analysis time (more than 6 h [12]) and, second, by the need to remove the spent active film after each analysis. A biological sensor based on a piezoelectric resonator with a lateral electric field allows bacteria to be analyzed directly in the liquid phase without an active layer for several minutes. The team of authors of this paper has previously shown the possibility of using a resonator with a lateral electric field based on lithium niobate to record various biological interactions of bacterial cells with antibodies, bacteriophages, and mini-antibodies [13,14,15] with the sensitivity threshold of 10^4^ cells/mL. However, the quality factor of the pointed resonator on the plate of lithium niobate in contact with the liquid was found to be too low [26], since the excited longitudinal wave in such a resonator led to radiation loss in the liquid. The sensor based on a resonator made of PZT ceramics presented in this paper allowed a reduction in the value of the threshold up to 10^3^ cells/mL. This may be explained by the fact that for the dominant mode of the PZT resonator the shear component of the mechanical displacement is predominant and this greatly reduces radiation loss upon contact with the liquid. It has been also shown that the sensor confidently detects specific biological interactions of microbial cells in aqueous solutions with conductivity up to 300 μS/cm. Such conductivity is characteristic, for example, of well-treated tap water.

The results obtained showed that the magnitude of the change in the output parameters of the investigated sensor depends on the concentration of bacterial cells in the cell suspension. This opens up the prospects for further work in terms of quantitative analysis of microbial cells. Furthermore, we plan to continue research on this sensor in order to study its capabilities for rapid analysis of viral particles.

## 4. Conclusions

The conducted studies showed that the interaction of microbial cells with specific antibodies leads to a significant change in the electrical impedance of the sensor based on a resonator with a lateral electric field made of PZT ceramics. The sensor has high sensitivity and speed. It was shown that the sensor confidently detects microbial cells with a low concentration (10^3^ cells/mL) not only in distilled water, but also in buffer solutions with conductivity as high as 300 μS/cm. This opens up the possibility of using a sensor for the analysis of microorganisms in real liquids, such as tap and industrial water, and food liquids. The absence of a sensor reaction to a non-specific biological interaction allows the use of this sensor not only to detect bacterial cells, but also to identify them in an aquatic environment. A short analysis time, which did not exceed 4 min in this study, allows the use of this sensor for rapid diagnosis of microbial cells.

## Figures and Tables

**Figure 1 sensors-20-03003-f001:**
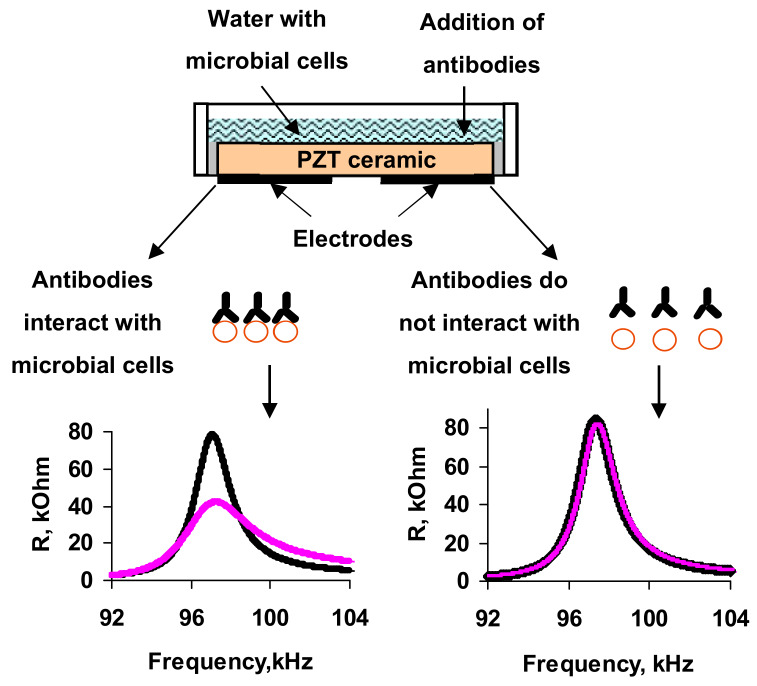
The general scheme of the experiment.

**Figure 2 sensors-20-03003-f002:**
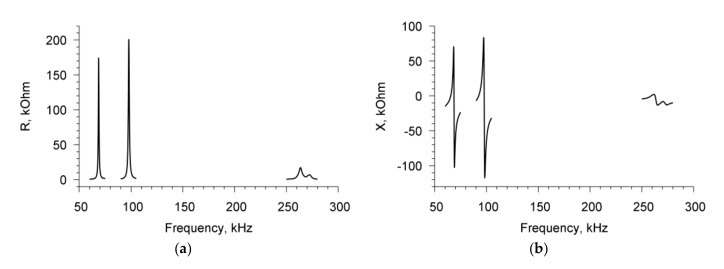
The frequency dependencies of the real (**a**) and imaginary (**b**) parts of the electrical impedance of PZT resonator with empty liquid container.

**Figure 3 sensors-20-03003-f003:**
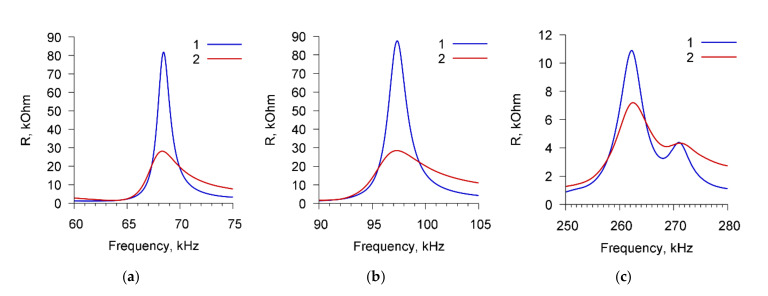
The frequency dependencies of the real part of the electrical impedance of PZT resonator with container loaded by *A. brasilense* Sp7 suspension before (curve 1) and after (curve 2) adding the specific Abs. The cell concentration was 10^3^ cells/mL, the concentration of Abs was 4 μg/mL. (**a**) Peak near 68.7 kHz, (**b**) peak near 97.8 kHz, (**c**) peak near 264 kHz.

**Figure 4 sensors-20-03003-f004:**
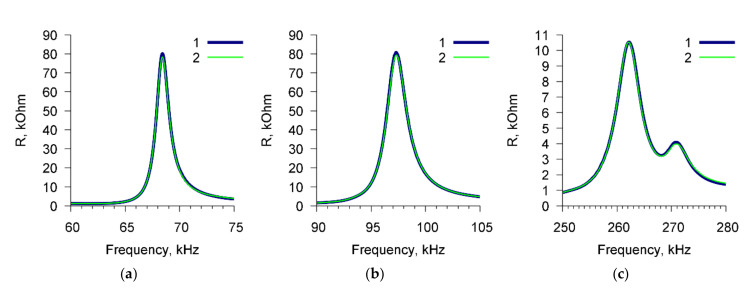
The frequency dependencies of the real part of the electrical impedance of PZT resonator with container loaded by *E. coli* K-12 suspension before (curve 1) and after (curve 2) adding the Abs specific to *A. brasilense* Sp7. The cell concentration was 10^3^ cells/mL, the concentration of Abs was 4 μg/mL. (**a**) Peak near 68.7 kHz, (**b**) peak near 97.8 kHz, (**c**) peak near 264 kHz.

**Figure 5 sensors-20-03003-f005:**
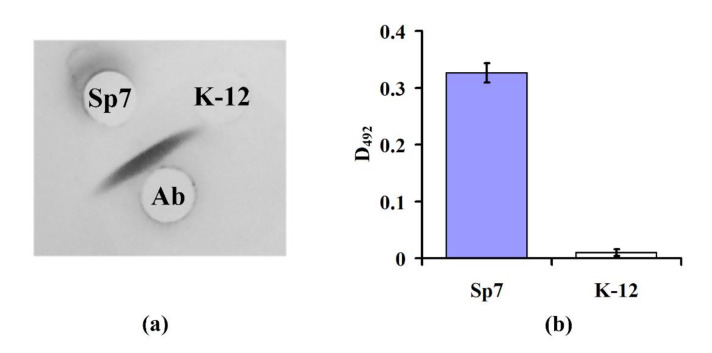
Results of the immunodiffusion assay (**a**) and ELISA (**b**) of strains *A. brasilense* Sp7 and *E. coli* K-12 with the antibodies against *A. brasilense* Sp7 O-antigen (Ab). For ELISA, the optical density of the enzyme reaction for a concentration of 10^6^ cells/mL is shown.

**Figure 6 sensors-20-03003-f006:**
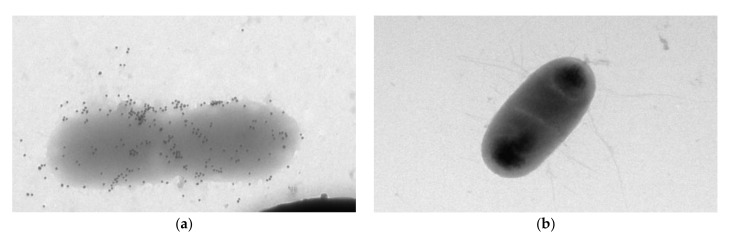
Electron microscopy of (**a**) cells of *A. brasilense* Sp7 with Abs specific to *A. brasilense* Sp7 labeled with colloidal gold (× 10,000) and (**b**) *E. coli* K-12 with Abs specific to *A. brasilense* Sp7.

**Figure 7 sensors-20-03003-f007:**
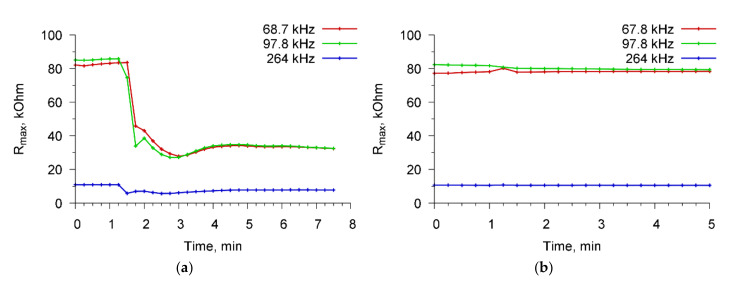
The time dependencies of the maximum of real part of the electrical impedance of PZT resonator during a specific interaction “*A. brasilense* Sp7—specific Abs” (**a**) and nonspecific interaction “*E. coli* K-12—Abs specific to *A. brasilense* Sp7 (**b**), for resonance peaks near frequencies 68.7, 97.8, and 264 kHz.

**Figure 8 sensors-20-03003-f008:**
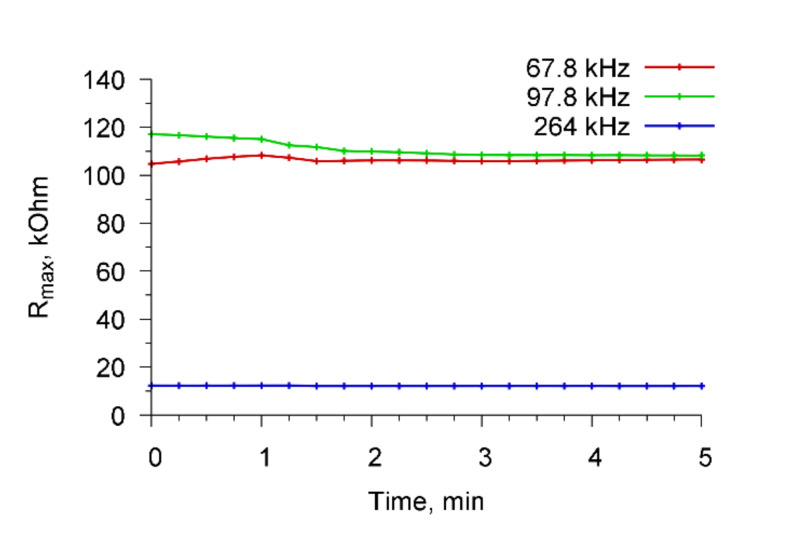
The time dependencies of the maximum of the real part of the electrical impedance of PZT resonator at addition of Abs specific to *A. brasilense* Sp7 to distilled water without cells for resonance peaks near the frequencies 68.7, 97.8, and 264 kHz.

**Figure 9 sensors-20-03003-f009:**
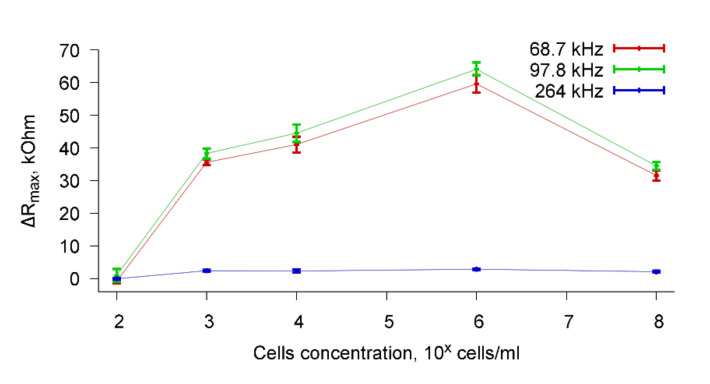
Dependencies of ΔR_max_ of the resonator on the concentration of the *A. brasilense* Sp7 cells for resonance peaks near the frequencies 68.7, 97.8, and 264 kHz where x is power. The concentration of antibodies was 4 μg/mL.

**Figure 10 sensors-20-03003-f010:**
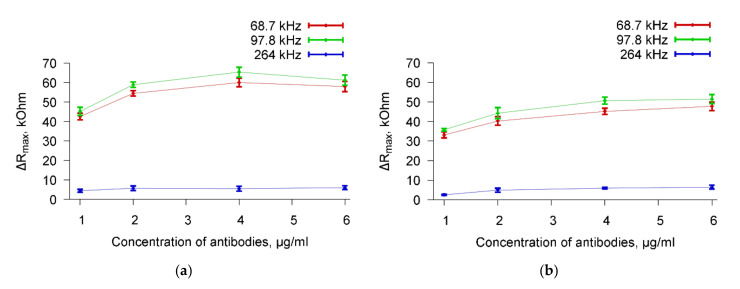
Dependencies of the change of ΔR_max_ of the resonator on the Abs concentration for *A. brasilense* Sp7 cell suspension after adding the specific Abs for resonance peaks near frequencies 68.7, 97.8, and 264 kHz. The cell concentration is 10^3^ (**a**) and 10^8^ (**b**) cells/mL.

**Figure 11 sensors-20-03003-f011:**
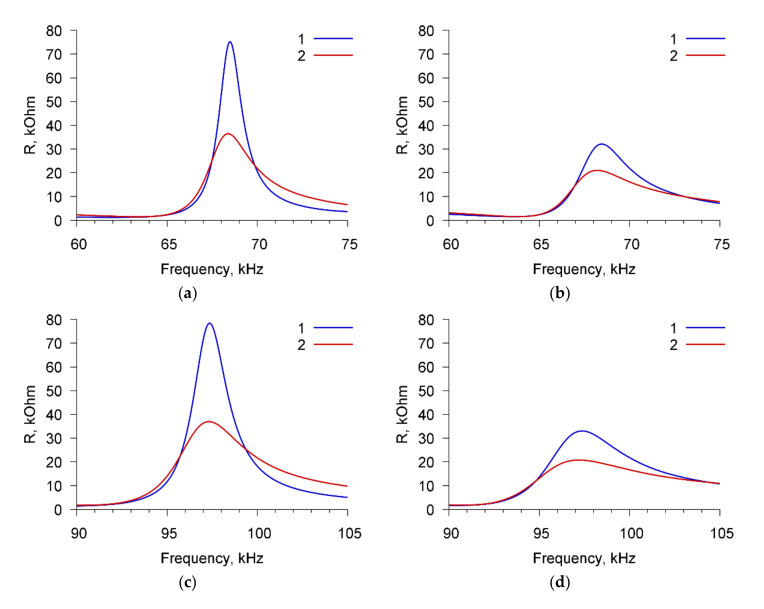
The frequency dependencies of the real part of the electrical impedance of a PZT resonator with a container loaded with *A. brasilense* Sp7 suspension before (curve 1) and after (curve 2) adding the specific Abs. The cell concentration was 10^3^ cells/mL, the concentration of antibodies was 6 μg/mL. The buffer conductivity was 10 μS/cm (**a**, **c**, **e**) and 50 μS/cm (**b**, **d**, **f**). The resonance peaks correspond to frequencies: 68.7 kHz (**a**, **b**), 97.8 kHz (**c**, **d**), and 264 kHz (**e**, **f**).

**Figure 12 sensors-20-03003-f012:**
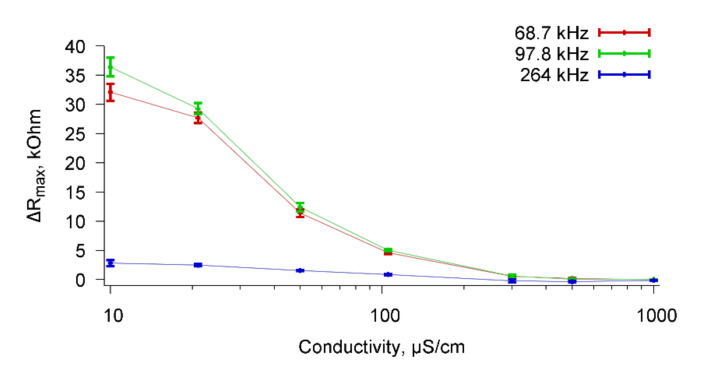
Dependencies of ΔR_max_ on the conductivity of the buffer solution when specific Abs was added to the buffer with the cells *A. brasilense* Sp7 for resonance peaks near frequencies 67.8, 98.7, and 264 kHz.

**Figure 13 sensors-20-03003-f013:**
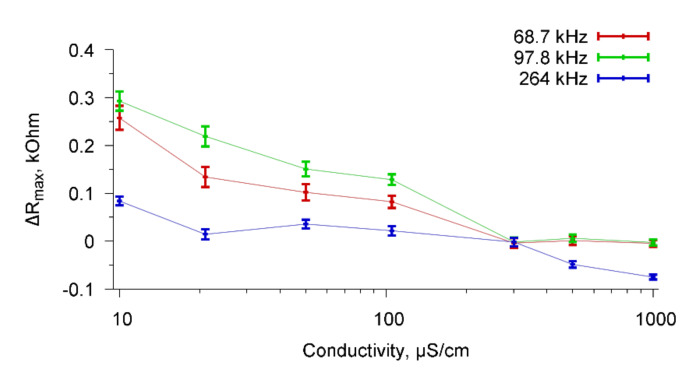
Dependencies of ΔR_max_ on the conductivity of the buffer solution when Abs specific for *A. brasilense* Sp7 cells was added to the buffer with *E. coli* K-12 cells for resonance peaks near frequencies 67.8, 98.7, and 264 kHz.

**Table 1 sensors-20-03003-t001:** The values of ΔR_max_ for suspensions of *A. brasilense* Sp7 cells with a concentration of 10^3^ cells/mL based on distilled water (4.5 μS/cm) and buffer solutions (10–1000 μS/cm) due to the addition of specific Abs in an amount of 6 μg/mL for resonant peaks near frequencies of 67.8, 98.7, and 264 kHz.

Conductivity, μS/cm	ΔR_max_, kOhm, peak 68.7 kHz	ΔR_max_, kOhm, peak 97.8 kHz	ΔR_max_, kOhm, peak 264 kHz
4.5	63.5303	65.5608	4.09908
10	32.0923	36.3881	2.81509
20	27.6936	29.2813	2.48143
50	11.3743	12.3867	1.53813
100	4.59971	5.02662	0.86089
300	0.64757	0.68282	0.32149
500	0.17027	0.25516	0.13604
1000	0.09172	0.03193	0.04684
500 (pure water)	0.18135	0.26791	0.12341
1000 (tap water)	0.09163	0.03152	0.03571

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
