# Peer review of "Sensor Based on PZT Ceramic Resonator with Lateral Electric Field for Immunodetectionof Bacteria in the Conducting Aquatic Environment ††"

_sensors, 2020, doi:10.3390/s20103003_

Round 1
Reviewer 1 Report
The work from Borodina et al report on the utilization of a PZT ceramic resonator for the detection of model bacteria in laboratory samples with different conductivity.
The work is interesting and the scientific methodology adequate. Nevertheless, few aspects need to be improved.
1) One step seems to be missing in the ELISA description: recognition of the bacteria with the specific antibodies
2) Figure 5a should be described a bit more in details to help reader not expert of the technique to understand the results.
3) In Figure 10 the effect of Abs concentration on the response (for 10^3 cell/ml) seems to be minor; is this the same for the highest concentration (10^8 cell/ml)?
4) In all the Figure concerning the detection of the cells there are no error bars. Error bars should be added to provide information on (i) reproducibility if the measurements and (ii) meaningfulness of the observations.
5) Authors should attempt detection of bacteria in tap water to show the applicability of their approach.
6) Authors quote few publications on the association of changes in conductivity with antibody bacteria interaction. In order to help the reader, the authors should describe in the text the proposed mechanism for these changes.
7) Discussion part should discuss results, underlining mechanisms and possible future application but not setting the motivation of the work (that is mainly introduction topic).
Author Response
The work from Borodina et al report on the utilization of a PZT ceramic resonator for the detection of model bacteria in laboratory samples with different conductivity.
The work is interesting and the scientific methodology adequate. Nevertheless, few aspects need to be improved.
- One step seems to be missing in the ELISA description: recognition of the bacteria with the specific antibodies
Thank you for this comment, we have introduced the following text in the lines 81-84
This solution was replaced by 50 μl of Abs specific to the lipopolysaccharide of A. brasilense Sp7 (final conc. 0.2 mg/ml) diluted in phosphate-buffered saline (PBS) with 0.02% Tween 20 and 0.005% PEG (for prevention of nonspecific Ab sorption). After incubation for 30 min, the wells were washed three times with 100 μl of PBS–0.02% Tween 20.
- Figure 5a should be described a bit more in details to help reader not expert of the technique to understand the results.
Thank you for this comment. We have corrected and expanded the text in the lines 217-227:
During double immunodiffusion, specific antigen and antibody placed into wells of the agar gel diffuse towards each other and they form the precipitation in the form of a band at the zone of their meeting. According to the results of immunodiffusion, one can clearly see (Figure 5a) that Abs interacted only with the extract of A. brasilense Sp7 cells (formation of the antigen-antibody complex immunoprecipitation as the band in agarose gel) and did not interact with the extract of E. coli K-12 (no immunoprecipitation band). So, the Abs against O antigen of A. brasilense Sp7 specifically formed the insoluble antigen-antibody complex only with the surface antigens of A. brasilense Sp7 [24] and did not form it with antigens of E. coli K-12. ELISA is characterized by the unique specificity of the immunochemical reaction, i.e. antibodies bind exclusively to certain antigens and are highly sensitive to antigen detection. The direct specificity of the Abs used in the work was also determined by ELISA.
- In Figure 10 the effect of Abs concentration on the response (for 10^3 cell/ml) seems to be minor; is this the same for the highest concentration (10^8 cell/ml)?
We have added the Figure 10b for the cell concentration of 10^8 cell/ml. Correspondingly, we have corrected the text below the Figure10. Instead the text in lines 312-318:
It can be seen that the addition of specific Abs to the cell suspension leads to a significant change in Rmax even with their minimal amount. For an antibody concentration of 1 μg/ml, this change was equal to 42, 45, and 4.5 kOhm for resonances near the frequencies of 68.7, 97.8, and 264 kHz, respectively. A further increase in the Abs concentration (from 2 to 6 μg/ml) led to an even larger change in Rmax. The maximum value of ∆Rmax turned out to be 57, 65, and 6 kOhm for resonances near 68.7, 97.8, and 264 kHz, respectively. Similar dependencies were constructed for other concentrations of microbial cells.
we have included in lines 319-330 the following text:
Figure 10a shows, that adding 1 μg/ml of antibodies to a cell suspension with a concentration of 103 cells/ml leads to a change in Rmax (∆Rmax) of 42, 45, and 4.5 kOhm for resonances near the frequencies of 68.7, 97.8, and 264 kHz, respectively. The similar ∆Rmax values are equal to 57, 65, and 6 kOhm for the amount of antibodies of 5 μg/ml. Figure 10b presents the same data for a cell suspension with a concentration of 108 cells/ml. In this case, the ∆Rmax values turns out to be 32, 35, and 3 kOhm (the number of antibodies = 1 μg/ml) and 45, 50, and 5 kOhm (the number of antibodies = 5 μg/ml) for resonances near 68.7, 97.8, and 264 kHz, respectively. One can see a decrease in the sensor response for the concentration of 108 cells/ml as compared to the concentration of 103 cells/ml. An explanation of this fact is given after the commentary on Figure 9.
So, it has been shown that the addition of specific Abs to the cell suspension leads to a significant change in Rmax even with their minimal amount. Similar dependencies were constructed for other concentrations of microbial cells
4) In all the Figure concerning the detection of the cells there are no error bars. Error bars should be added to provide information on (i) reproducibility if the measurements and (ii) meaningfulness of the observations.
Thank you for this comment. We have added the Error bars in Figures 9, 10, 12, and 13. The graphs of the frequency dependencies in Figures 3, 4 and 11 containing more than 1000 frequency points are given without the error bars, because in the presented scale they will be simply invisible.
5) Authors should attempt detection of bacteria in tap water to show the applicability of their approach.
Thank you for this comment. We carried out additional measurements in tap and drinking water.
From https://www.lenntech.com/applications/ultrapure/conductivity/water-conductivity.htm
we have found the following information:
Typical conductivity of waters:
Ultra pure water: 5.5×10-6 S/m = 5.5×10-2 μS/cm.
Drinking water: 0.005 – 0.05 S/m = 50 – 500 μS/cm.
Tap water: ~1000 μS/cm.
The measurement results were added in the lower two rows of Table 1.
We have added the following text in lines 389-392:
Additionally, the measurements were carried out with the samples of tap water (the conductivity = 1000 μS/cm), as well as with the samples of purified drinking water (the conductivity = 500 μS/cm). The data obtained are presented in the lower two rows of Table 1. They are well comparable with the data for buffer solutions
6) Authors quote few publications on the association of changes in conductivity with antibody bacteria interaction. In order to help the reader, the authors should describe in the text the proposed mechanism for these changes.
Thank you for this comment. We have included in the lines 189-203 the following text:
The parameters of the sensor changed due to an increase in the conductivity of the cell suspension caused by a specific biological interaction. This can be explained as follows. When specific antibodies are added to the cell suspension, the active center of the antibody (Ab) joins with the antigenic (Ag) determinant. This highly specific process proceeds in aqueous solutions at a high speed. O-antigen of the gram-negative bacteria is associated with LPS of the cell wall. At that the determinant groups of this complex antigen are the terminal repeating units of polysaccharide chains attached to its main part. At the moment of binding the antibody to the determinant group of the antigen, a conformational rearrangement of the active center of the antibody occurs and the active antibody adapts to the antigen. The binding of antibodies to antigenic determinants on the cell surface is recorded by changing the polarization properties of the Ag-Ab complex. The interaction of an antigen molecule with an antibody or its active Fab fragment is accompanied, in turn, by changes in the spatial structure of the antigen molecule. This affects the change in the integrity of bacterial membranes, and leads to the release of cell contents into suspension and to the increase in its electrical conductivity. This increase in conductivity is recorded by the acoustic sensor as a decrease in the real and imaginary parts of the electrical impedance of the resonator near the resonant frequencies.
7) Discussion part should discuss results, underlining mechanisms and possible future application but not setting the motivation of the work (that is mainly introduction topic).
Thank you, we have removed the Discussion subsection from the section Results and Discussion with the references [25-33] and have included in lines 469-485 the following text:
It should be noted that in most works devoted to acoustic biological sensors [5, 8, 10], active layers with the immobilized antibodies are used. Such approach is distinguished, first, by the long analysis time (more than 6 hours [10]) and, second, by the need to remove the spent active film after each analysis. A biological sensor based on a piezoelectric resonator with a lateral electric field allows bacteria to be analyzed directly in the liquid phase without an active layer for several minutes. The team of the authors of this paper has previously shown the possibility of using a resonator with a lateral electric field based on lithium niobate to record various biological interactions of bacterial cells with antibodies, bacteriophages and mini-antibodies [11, 12, 13] with the sensitivity threshold of 104 cells/ml. But the quality factor of the pointed resonator on the plate of lithium niobate in contact with the liquid turned out too low [23], since the excited longitudinal wave in such a resonator led to the radiation loss in the liquid. The sensor based on a resonator made of PZT ceramics presented in this paper allowed to decrease the value of the threshold up to 103 cells/ml. This may be explained by the fact that for the dominant mode of the PZT resonator the shear component of the mechanical displacement is predominant and this greatly reduces radiation loss upon contact with the liquid. It has been also shown that the sensor confidently detects specific biological interactions of microbial cells in aqueous solutions with conductivity up to 300 μS/cm. Such conductivity is characteristic, for example, of well-treated tap water.

Reviewer 2 Report
The article deals with the development of a sensor based on PZT ceramics resonator with lateral electric field for bacterial immunodetection in conducting aquatic environment. The idea is interesting and presents innovation, the article is well conducted and is written in good English, explaining clearly the steps of sensor construction, characterization and the results obtained. I consider that the article is appropriate for the publication in Sensors; however, I recommend small changes, as follows:
Introduction, line 38-39: “Among a large number of acoustic liquid and biological sensors, based on piezoelectric resonators with a lateral electric field are in a special position” seems like some word is missing…maybe “Among a large number of acoustic liquid and biological sensors, those based on piezoelectric resonators with a lateral electric field are in a special position”.
Introduction, line 55: A “.” is missing at the end of the sentence, after references.
Subsection 2.2, line 87: “bacterial cells were using as the control”, should be “bacterial cells were used as the control”
Subsection 2.4, line 95: Please, check the subscripts of the chemical formula.
Subsection 3.1, line 152: “in comparison with the adding the water without cells” should be “in comparison with adding the water without cells”
Page 9, line 261: “leads to a some decrease in…” should be “leads to some decrease in…”
Page 9, line 269: “At that the addition of…” should be “However, the addition of…”
Figure 9, 10, 12 and 13: I suggest representation by scatter instead of line or maybe scatter with lines, but thinner lines only as guidelines. Because there are not enough points to express that exact trend.
Page 10, line 316: Please substitute the Russian character.
Page 13, line 383: Please substitute the “.” Between references with “,”
General remarks
The Discussion section is rather strange, I am not sure makes sense introducing this section. I think there is information in there that should go to Introduction (most of it); there is also some repetition with Introduction.
I suggest a comparison with literature for similar sensors, if any, as regarding particularly the lower limit of determination. The only comparison in the text is with studies of the same authors, there are no other studies, beside these?
Author Response
Open Review #2
The article deals with the development of a sensor based on PZT ceramics resonator with lateral electric field for bacterial immunodetection in conducting aquatic environment. The idea is interesting and presents innovation, the article is well conducted and is written in good English, explaining clearly the steps of sensor construction, characterization and the results obtained. I consider that the article is appropriate for the publication in Sensors; however, I recommend small changes, as follows:
Introduction, line 38-39: “Among a large number of acoustic liquid and biological sensors, based on piezoelectric resonators with a lateral electric field are in a special position” seems like some word is missing…maybe “Among a large number of acoustic liquid and biological sensors, those based on piezoelectric resonators with a lateral electric field are in a special position”.
Thank you for this comment, we have corrected the sentence in lines 38-40:
Among a large number of acoustic liquid and biological sensors, those based on piezoelectric resonators with a lateral electric field are in a special position.
Introduction, line 55: A “.” is missing at the end of the sentence, after references.
Thank you, we have added the dot in the line 55.
Subsection 2.2, line 87: “bacterial cells were using as the control”, should be “bacterial cells were used as the control”
Instead “were using” we have included “were used” in line 90.
Subsection 2.4, line 95: Please, check the subscripts of the chemical formula.
Thank you, we have corrected the subscripts of the chemical formula in lines 99-100:
0.12 M phosphate-buffered saline (PBS) (pH 7.2) of the following composition (g/l): KH2PO4, 0.43; Na2HPO4, 1.68; NaCl, 7.2.
Subsection 3.1, line 152: “in comparison with the adding the water without cells” should be “in comparison with adding the water without cells”
We have corrected the sentence in lines 157-158 (have removed “the”).
Page 9, line 261: “leads to a some decrease in…” should be “leads to some decrease in…”
We have corrected the sentence in line 288 (have removed “a”).
Page 9, line 269: “At that the addition of…” should be “However, the addition of…”
We have corrected the sentence in line 296 (instead “At that” we have included “However,”).
Figure 9, 10, 12 and 13: I suggest representation by scatter instead of line or maybe scatter with lines, but thinner lines only as guidelines. Because there are not enough points to express that exact trend.
We agree with this remark. We have changed the presentation of Figures 9, 10, 12, and 13. We have added a measurement error bars to them, and made the connecting lines more thin. If the lines are completely removed, then the perception of the data presented in the Figures is difficult.
Page 10, line 316: Please substitute the Russian character.
We have changed Russian “и” by '' and'' in line 355.
Page 13, line 383: Please substitute the “.” Between references with “,”
We have removed this text from the manuscript.
General remarks
The Discussion section is rather strange, I am not sure makes sense introducing this section. I think there is information in there that should go to Introduction (most of it); there is also some repetition with Introduction.
Thank you, we have removed the Discussion subsection from the section Results and Discussion with references [25-23] and have included in lines 469-485 the following text:
It should be noted that in most works devoted to acoustic biological sensors [5, 8, 10], active layers with the immobilized antibodies are used. Such approach is distinguished, first, by the long analysis time (more than 6 hours [10]) and, second, by the need to remove the spent active film after each analysis. A biological sensor based on a piezoelectric resonator with a lateral electric field allows bacteria to be analyzed directly in the liquid phase without an active layer for several minutes. The team of the authors of this paper has previously shown the possibility of using a resonator with a lateral electric field based on lithium niobate to record various biological interactions of bacterial cells with antibodies, bacteriophages and mini-antibodies [11, 12, 13] with the sensitivity threshold of 104 cells/ml. But the quality factor of the pointed resonator on the plate of lithium niobate in contact with the liquid turned out too low [23], since the excited longitudinal wave in such a resonator led to the radiation loss in the liquid. The sensor based on a resonator made of PZT ceramics presented in this paper allowed to decrease the value of the threshold up to 103 cells/ml. This may be explained by the fact that for the dominant mode of the PZT resonator the shear component of the mechanical displacement is predominant and this greatly reduces radiation loss upon contact with the liquid. It has been also shown that the sensor confidently detects specific biological interactions of microbial cells in aqueous solutions with conductivity up to 300 μS/cm. Such conductivity is characteristic, for example, of well-treated tap water.

Round 2
Reviewer 1 Report
The reviewer thanks the authors for the revisions and consider the manuscript adequate for publication.